# Challenges and Solutions to Viral Diseases of Finfish in Marine Aquaculture

**DOI:** 10.3390/pathogens10060673

**Published:** 2021-05-30

**Authors:** Kizito K. Mugimba, Denis K. Byarugaba, Stephen Mutoloki, Øystein Evensen, Hetron M. Munang’andu

**Affiliations:** 1Department of Biotechnical and Diagnostic Sciences, College of Veterinary Medicine Animal Resources and Biosecurity, Makerere University, Kampala P.O. Box 7062, Uganda; dkb@covab.mak.ac.ug; 2Department of Paraclinical Sciences, Faculty of Veterinary Medicine, Norwegian University of Life Sciences, P.O. Box 369, 0102 Oslo, Norway; stephen.mutoloki@nmbu.no (S.M.); oystein.evensen@nmbu.no (Ø.E.); 3Department of Production Animal Clinical Sciences, Faculty of Veterinary Medicine, Norwegian University of Life Sciences, P.O. Box 369, 0102 Oslo, Norway

**Keywords:** viruses, challenges, solutions, prevention, control, vaccines, biosecurity

## Abstract

Aquaculture is the fastest food-producing sector in the world, accounting for one-third of global food production. As is the case with all intensive farming systems, increase in infectious diseases has adversely impacted the growth of marine fish farming worldwide. Viral diseases cause high economic losses in marine aquaculture. We provide an overview of the major challenges limiting the control and prevention of viral diseases in marine fish farming, as well as highlight potential solutions. The major challenges include increase in the number of emerging viral diseases, wild reservoirs, migratory species, anthropogenic activities, limitations in diagnostic tools and expertise, transportation of virus contaminated ballast water, and international trade. The proposed solutions to these problems include developing biosecurity policies at global and national levels, implementation of biosecurity measures, vaccine development, use of antiviral drugs and probiotics to combat viral infections, selective breeding of disease-resistant fish, use of improved diagnostic tools, disease surveillance, as well as promoting the use of good husbandry and management practices. A multifaceted approach combining several control strategies would provide more effective long-lasting solutions to reduction in viral infections in marine aquaculture than using a single disease control approach like vaccination alone.

## 1. Introduction

Aquaculture, also referred to as “underwater agriculture”, is the fastest growing food-producing sector in the world. By 2013, it had already overtaken beef and poultry production [1]. It contributes one-third of global food production, which includes more than 200 farmed fish species [1]. Similar to other intensive farming systems, the impact of infectious diseases in intensive fish farming has had serious repercussions. Among these are viral diseases, most of which cause high economic losses [2,3,4]. In addition, viral diseases reduce fish welfare by causing various conditions that adversely affect the wellbeing of fish, such as reduced feed intake, abnormal swimming behavior, predation of diseased fish, and negative social interactions [5,6,7,8]. As pointed out by several scientists [9,10,11,12], viruses are the most abundant entity in marine ecosystems. Their abundance in oceans range from 3 × 10^6^ mL^−1^ viral particles (VPs) in the deep sea to 1 × 10^8^ mL^−1^ VPs in coastal waters, decreasing in abundance with distance from shore [9,13,14]. However, only a few viruses cause diseases in farmed fish linked to high economic losses, rendering these viruses noticeable in most countries. It should be noted, however, that most pathogenic viruses of fish have been detected in areas around Europe, Americas, and Asia. Hence, it is likely that several pathogenic viruses remain unknown due to lack of surveillance. In this review, we highlight some of the major challenges associated with viral diseases in mariculture and suggest solutions to these problems.

## 2. Major Challenges Associated with Viral Diseases in Marine Aquaculture

### 2.1. Emerging Infectious Diseases

Krkosek [15] defined emerging infectious diseases (EID) as previously unknown diseases or spread of an existing disease into a new host or geographic area. As shown in Figure 1, factors associated with occurrence of EIDs include (i) viral factors such as the emergence of a new virus in an area or viruses acquiring virulence through mutations, (ii) anthropogenic activities such as the introduction of an exotic susceptible fish species or virus in a new habitat, (iii) host factors such as fish becoming stressed and thus susceptible to viral infection through intensive aquaculture, or (iv) environmental factors such as changes in salinity, pH, CO_2_, and other factors that render fish susceptible to infection [16]. Krkosek [15] pointed out that freshwater and marine fish have the highest rates of EIDs among vertebrates mainly because they are the most speciose group [17]. This could also be attributed to the fact that aquatic environments used for fish farming have a high population of unknown viruses, some of which could infect and cause disease in fish. During the period of 1988 to 1992, Hetrick and Hedrick [18] identified more than 35 EIDs that infect fish. Among these were new infectious agents, while others were isolates of known agents but infecting new host species. They attributed this increase in EIDs to an increase in surveillance accompanied by an expansion of aquaculture that included new farmed fish species. Additionally, we reported close to 20 newly discovered viral pathogens of fish using metagenomics analysis between 2010 and 2016 [19]. Overall, EIDs pose a challenge to the expansion of aquaculture.

### 2.2. Migratory Species

Migratory fish species have been reported to spread viral pathogens over long distances. One of the factors that favor viral spread in aquatic environments is that ocean environments have fewer barriers, enabling ocean currents to carry both hosts and pathogens over long distances [20]. In addition, behavioral factors such as the formation of schools and shoals in migratory species facilitate viral transmission over long distances. These factors cause viral spread to be at a higher rate in oceans than on-land transmission in higher vertebrates [20]. For example, the herpesvirus epidemic that caused massive mortalities of pilchards (*Sardinops sagax*) along a 5000 km distance of the Australian and 500 km of the New Zealand coastlines spread at a rate of approximately 30 km/day in relation to the migration rate of pilchards supported by prevailing sea currents in 1995 [21,22,23,24]. Similarly, herring off the Scotland coastal areas have been reported to carry Baltic strains of viral hemorrhagic septicemia virus (VHSV) over long distances [25].

Another important factor associated with viral spread is the return of migratory species to spawning sites. Ferguson et al. [26] noted that returning migrating steelhead trout (*Oncorhynchus mykiss*) were a vital source of infectious hematopoietic necrosis (IHNV) infections to farmed fish in the Columbian river basin in the USA, cases of which were high during the spawning of wild trout. Similarly, Traxler et al. [27] reported the occurrence of IHNV infections in sockeye salmon (*O. nerka*) during spawning migrations in British Colombia in Canada, while Nylund et al. [28] observed an association between wild herring (*Chipea harengus*) migration and ISAV outbreaks on fish farms in Norway and Canada. Haenen et al. [29] observed that eels migrate over long distances to their spawning sites. The European eel (*Anguilla anguilla*) travels over 5500 km to the Sargasso Sea [30,31,32], while the American eel (*A. rostrata*) migrates over 4000 km to the Sargasso Sea [31,32]); the Australian eel (*A. australis*) covers 5000 km to the Pacific Ocean [33], and the Japanese eel (*A. japonica*) covers 4000 km to the Marianna Islands in the Philippines to spawn [34]. Viruses isolated from migrating eels include herpesviruses [35], rhabdoviruses [36], and infectious pancreatic necrosis virus (IPNV) [37] which could be transmitted over long distances.

### 2.3. Anthropogenic Activities

Human activities have been shown to play a vital role in the spread of viral diseases of farmed fish. In Norway, expansion of the Atlantic salmon industry has been linked with the spread of piscine orthoreovirus (PRV), piscine myocarditis virus (PMCV), IPNV, and salmon anemia virus (SAV) into marine environments [38]. In Chile, Mardones et al. [39] showed that anthropogenic activities contributed to the spread of ISAV during the 2007 to 2009 outbreaks. Major contributing factors included the movement of harvested live fish or fish byproducts between farms. They observed that management factors such as the coexistence of multiple fish generations contributed to an increase in ISAV cases on the farms. They also noted that an increase in the number of ships entering a farm contributed to an increase in ISAV outbreaks. Further, they observed that shorter distances between farms were associated with reduced time to infection, while farms located further apart had a longer time to infection.

Industrial activities such as mining and construction have been associated with the spread of viral diseases in the oceans. For example, the construction of the Suez Canal led to the introduction of close to 41 Indo-Pacific fish species into the Mediterranean Sea, which contributed to the spread of viral disease in the area [40]. Lampert et al. [41] observed that nervous necrosis virus (NNV) was significantly higher in Nemipterus randalli, which is a relatively newly established invasive Suez Canal species in the Mediterranean Sea, than in Sardinella aurita and Lithognathus mormyrus that are indigenous Mediterranean Sea species. This finding shows that an exotic invasive fish species introduced in a new area could play a vital role in amplifying and increasing the release of viruses in the environment to higher levels than native species.

Detection of viruses such as ranavirus, lymphocystis virus (LCDV), VHSV, NNV, SAV, ISAV, PMCV, PRV, and IPNV [42,43,44,45,46,47,48,49] from wrasse (*Labridae*) species used for sea lice control suggests that human intervention using cleaner fish to control sea lice on Atlantic salmon could be transmitting pathogenic viruses to farmed fish. The use of virus-infected fish as a protein source for farmed fish is another human activity that poses a danger of transmitting pathogenic viruses to farmed fish. For example, the use of marine fish as feed is believed to have been responsible for introducing VSHV to rainbow trout in freshwater in Europe [50], while imported frozen fish fed to tuna may have introduced pilchards’ herpesvirus to wild pilchards in Australia, resulting in mass mortalities [51]. Similarly, wet feed produced from herring contaminated with ISAV could have transmitted the virus to Atlantic salmon [28]. Altogether, these studies show that various anthropogenic activities pose a danger, spreading viral diseases in aquaculture.

### 2.4. Wild Reservoirs of Marine Viruses

Giacopello et al. [52] detected NNV in healthy wild cardinal fish (*Pterapogon kauderni*), spiny eel (*Macrognathus aculeatus*), and slimehead (*Hoplostethus mediterraneus*) at demersal level below 300 m depth [52,53], pointing to the existence of VERV in the deep sea below levels used for aquaculture. Similarly, Wallace et al. [41] identified several fish species that included the common dab (*Limanda limanda*), plaice (*Pleuronectes platessa*), lemon sole (*Microstomus kitt*), flounder (*Platichthys flesus*), and long rough dab (*Hippoglossoides platessoides*) as wild reservoirs of IPNV in Scottish marine water using demersal trawling. Berzak et al. [54] detected NNV in different wild fish species sampled at different trophic levels, including demersal levels in the Levantine Basin of the Mediterranean Sea. They also detected NNV in farmed gilthead sea bream (*Sparus aurata*) after 120 days of culture in the same area, pointing to the transmission of NNV from wild fish at demersal levels to farmed fish. Altogether, these studies show the existence of wild reservoirs of viral pathogens for farmed fish in deep-sea environments.

Estuaries, lagoons, fjords, and peninsulas that serve as transitional areas between fresh and marine water serve as important nurseries and breeding areas that stock different wild fish species sharing similar habitat properties [55]. These areas also serve as ideal places for fish farming. As such, they are bound to serve as viral transmission sites between farmed and wild fish. Snow et al. [56] detected SAV in wild common dab (*Limanda limanda*), long rough dab (*Hippoglossoides platessoides*), and plaice (*Pleuronectes platessa*) in fish farming areas in the Stonehaven Bay, Scotland. Similarly, the Skagerrak and Kattegat estuaries used for salmon and cod farming are habitats for several wild fish species that include herring, gobies (*Rhinogobius duospilus*), sprat (*Sprattus sprattus*), pipe fish (*Syngnathinae*), and European eel [55], of which viruses such as VHSV and IPNV have been detected from farmed and wild fish in the area [57]. The Gotland Gulf and Bothnian Bay in the Baltic Sea, also used for intensive farming of Atlantic salmon and cod, is a natural habitat for different wild fish species that include perch (*Perca fluviatilis*), burbot (*Lota lota*), herring (*Clupea harengus*), Roach (*Rutilus rutilus*), wild salmon, and cod [58]. Viruses such as VHSV and IPNV have been detected in wild and farmed fish in the area [58,59]. The Iberian Peninsula in the Gulf of Cadiz in Spain is a habitat for several wild fish species that include sardine (*Sardina pilchardus*), mackerel (*Scomber* spp.), common hake (*Merluccius merluccius*), and blue whiting (*Micromesistius poutassou*) [60]. The area is also used for the culture of various farmed fish species, such as sole (*Solea solea*) and Senegalese sole (*S. senegalensis*) [61]. Moreno et al. [62] detected IPNV and NNV in various wild and farmed fish species in the Gulf of Cadiz. In summary, these studies show that ecosystems that equally support the survival of farmed and wild fish can serve as spillover and spillback areas for viral transmission between farmed and wild fish [63].

Mao et al. [64] isolated identical iridoviruses from three-spined stickleback (*Gasterostelus aculeatus*) and red-legged frog (*Rana aurora*), showing that iridoviruses naturally infect animals belonging to different taxa. This finding was supported by Moody and Owens [65], who showed that injection or bath immersion of barramundi fish (*Lates calcarifer* Bloch) with Bohle iridovirus from the ornate burrowing frog (*Lymnodynastes ornatus*) resulted in a severe disease with high mortality. Altogether, these studies suggest that frogs may serve as a reservoir for fish viruses or vice versa.

### 2.5. Ballast Water

Ballast water has been regularly used in ships since the 1880s [66]. However, it poses the threat of transporting viruses between biomes. Leichsenring and Lawrence [67] estimated the viral particles (VPs) in ballast water during trans-Pacific voyages from South Korea to Vancouver to be 1.8 × 10^7^ VPs mL^−1^. They estimated the amount of ballast water discharged at >52 million tons every year in Canada. Taken together with Ruiz et al. [68], who estimated that each mL of ballast contains 7.4 × 10^6^ VPs, this implies that 3.9 × 10^20^ VPs are discharged in ballast water in Canada per annum. Drake et al. [69] estimated 6.8 × 10^19^ VPs discharged in ballast water in the Chesapeake Bay each year. Therefore, it is likely that similar VPs levels are discharged in different ports in the world.

Murray et al. [70] observed that the spread of ISAV to several Atlantic salmon farms across >850 km coastline in Scotland and the Orkney and Shetland islands in 1998–1999 was mainly associated with the movement of well boats that transported fish, and other supplies. They found a strong relationship between well boat visits and ISAV infections. Boats visiting an infected farm would fill up ISAV contaminated ballast water from areas around salmon cages. The ballast water in the boats was transported to other salmon farms, where it was discharged, releasing the virus in the environment. Similarly, it has been pointed out by several scientists [71,72] that VHSV could have been introduced into the Great lakes through ballast water, which resulted in mass mortalities of several fish species. This is supported by Sieracki et al. [72], who showed that fish farmed in locations that received ballast water from VHSV-infected sources in the Great Lakes were more likely to become infected. They observed that fish at ports that had the highest number of visits from VHSV-infected areas were more likely to be infected by VHSV, as shown in Montreal, which received the largest amount of ship traffic into the Great Lakes and had the highest rate of VHSV infection compared to other areas [73]. Altogether, these studies underscore the importance of ballast water in the spread of viral pathogens in marine aquaculture.

### 2.6. Limitations in Diagnostic Tools

The traditional approach of isolating viruses from infected tissues by cell culture followed by virus identification using different techniques is the most common approach for the diagnosis of fish viral diseases. This approach is useful in determining the causal-factor relationship between the disease and pathogen according to Koch’s postulates [74]. However, not all emerging viral pathogens have been cultured on cells. Some viruses such as PRV and PMCV have proved challenging and are yet to be cultured on cells. This hinders vaccine development by traditional approaches that require culturing, for example, inactivated viral vaccines. It suffices to point out that PCR has become the most widely used diagnostic tool for viral infections in marine aquaculture. One of the limitations with this technique is that it detects nucleic acids from viable and nonviable viruses. Hence, it has to be combined with other assays such as immunohistochemistry that link tissue damage with viral presence in infected organs. Other factors that can lead to false results using PCR include the use of degraded nucleic acids from poorly stored samples, presence of amplification inhibitors in the samples, and insufficient amount of detectable nucleic acids in the samples.

The limitations of diagnostic capabilities in poor-resource countries are a serious challenge because this derails the timely identification of causative agents of disease outbreaks. In some cases, it leads to sending samples from poor-resource countries to developed countries where appropriate diagnostic facilities are available. Maintaining the cold chain during the shipment of samples is another challenge, often leading to failure in isolating viruses from infected fish. Some countries do not have the appropriate resources to ensure that the virus in infected material is viable during transportation. Moreover, transportation of samples across different countries poses the danger of transboundary disease transmission. Further, the time lag between identification of the etiological agent and implementation of appropriate disease control measures exacerbates the spread of viral diseases. Other challenges include limitation in diagnostic tools able to determine viral mutations and failure to carry out disease surveillance.

### 2.7. International Trade

Fish and shellfish are among the most traded agricultural products in the world [75]. The trade of eggs, fingerlings, and adult fish has been linked to the transmission of viral disease across continents. In the 1950s, rainbow trout eggs were transported from the USA to Japan [76]. By 1955, the first outbreak of a severe infectious disease occurred in rainbow trout farms, but the pathogen was not identified until the 1960s [77]. Clinical signs and severe necrosis seen in the pancreas of infected fish were similar to those observed in trout in the USA [78]. The unknown disease was finally diagnosed as IPN by Sano [79]. Between 1967 and 1971, the Hokkaido Salmon Hatchery in Japan imported chinook salmon eggs from the USA. During the period of 1971–1972, an outbreak of IHNV was reported in sockeye salmon and kokanee salmon (*O. nerka*) hatched from eggs incubated together with imported eggs [80]. The source of the outbreak was traced to eggs imported from the USA.

Introduction of exotic species through trade has contributed to the spread of viral diseases to different continents. Vike et al. [81] noted that all salmonid species in the southern hemisphere were introduced from Europe and North America, which could have led to the spread of viral diseases accompanied with the introduction of salmonids in the southern hemisphere. They noted that the close relationship between ISAV strains from farmed Atlantic salmon in Chile and Norway points to a recent transmission of ISAV from Norway to Chile. Similarly, IPNV strains in Chile share genetic similarities with Norwegian strains [82], and detection of IPNV in Kenya [83] could be linked to the introduction of rainbow trout from the northern hemisphere. Although VHSV had been present in European waters for a long time, it is presumed that it did not cause disease outbreaks until rainbow trout was introduced for farming from North America [84]. Mohr et al. [85] and Rimmer et al. [86] isolated infectious spleen and kidney necrosis virus (ISKNV) from imported ornamental fish into Australia that led to ISKV outbreaks in farmed Platy (Xiphophorus maculatus). Similarly, Gomez et al. [87] detected NNV from imported fish into South Korea. In summary, these studies accentuate the role of international trade in the global transmission of viral disease in aquaculture.

## 3. Solutions to the Challenges Associated with Marine Viral Infection in Fish Aquaculture

### 3.1. Biosecurity

Moss et al. [88] defined biosecurity as the sum of all procedures put in place to protect farmed organisms from contracting, carrying, and spreading diseases. Implementation of biosecurity measures in aquaculture involves four management factors, namely, (i) fish, (ii) pathogens, (iii) environment, and (iv) personnel management. Implementation of biosecurity measures in fish management ensures that only healthy stocks are used for fish farming. This includes the screening of eggs for different pathogens and disinfection of eggs before incubation. Only eggs from hatcheries certified to be free of diseases should be used. All brood stocks should be screened for various diseases. All fish coming from outside should be sourced from certified disease-free farms and should be quarantined and screened for diseases on arrival. Inspection for any abnormal conditions should be carried out routinely. It is important to practice an all-in–all-out stocking to prevent introduction of stock that might be carrying diseases. Appropriate sanitation and disinfection should be practiced at all stages of the production cycle [89,90].

Implementation of biosecurity measures for pathogen management aims at preventing, reducing, or eliminating pathogens from stock, utensils, equipment, nets, tanks, cages, and other materials used for handling fish. Nets, buckets, and other fish handling equipment should be disinfected routinely. The efficacy of various disinfectants and antiseptics used in aquaculture has been reviewed by different scientists [91,92,93]. Environment biosecurity measures should include sanitation practices aimed at eliminating or reducing the existence of pathogens in places or facilities used for fish farming [91,94]. Implementation of biosecurity measures based on personnel management is only effective when personnel adhere to routine biosecurity practices and should start with management. Access should be limited to staff working in the fish facilities. Personnel coming from other fish facilities should be considered risk bearers. All staff should always put on protective clothing, while hand washing using antiseptics or disinfectants should be done routinely. Each staff member should be assigned to specific work areas. Health fish should be handled first before handling sick fish or quarantined fish.

### 3.2. Improved Diagnostic and Research Technologies

Diagnostic methods have evolved differently in various countries depending on the availability of reagents, cell lines for virus culture, laboratory equipment, and expertise to perform the diagnostic tests. Although traditional methods based on virus isolation using cell culture, histopathology, and in situ immunostaining of viral antigens account for the majority of diagnostics methods used in aquaculture, recent developments show that these methods are being taken over by molecular-biology-based techniques. As the cost of PCR-based detection methods becomes increasingly affordable in most countries, these methods are becoming routine diagnostic tools for viruses in aquaculture [95,96]. PCR-based assays have the advantage of producing rapid results. Other developments include the use of in situ hybridization (ISH) [97,98], which has the unique ability of detecting viral nucleic acid sequences in infected cells without altering cell morphology. This technique has contributed to elucidating cell tropism and localization of viral nucleic acids in different tissues [97,98]. On the other hand, quantitative PCR assays are widely used to quantify viral loads in infected tissues [99,100]. Combined application of these diagnostic tools has contributed to increasing our knowledge of host–pathogen interaction in viral infections of fish.

The major limitation of PCR-based diagnostic methods is that the design of primers used is based on prior knowledge of sequences of the virus under detection. Hence, PCR-based techniques lack the ability to detect unknown novel pathogens. However, the emergence of high-throughput sequencing (HTS) technologies such as metagenomics analysis having the ability to detect all nucleic acids present in a sample without prior knowledge of existing sequences has become a useful tool for the discovery of novel viruses infecting fish [101,102]. Metagenomics can be used to generate new sequences that do not share homology with sequences deposited in reference databases, thereby serving as a vital tool for novel viral discoveries [103,104]. In aquaculture, it has been used for the discovery of several novel viruses infecting fish and other aquatic organisms, antigenic variants, quasipecies, and viromes found in different ecosystems [105]. Moreover, several studies show that the duration between the first report of clinical disease and identification of the etiological agent is shorter using metagenomics analysis [19,106,107], unlike traditional diagnostic methods in which the duration from first report of clinical disease to identification of the etiological agent is longer. Therefore, metagenomics analysis is bound to expedite the discovery of novel pathogens in aquaculture. Moreover, it can be used to determine the viral community present in aquatic environments used for fish farming [108].

### 3.3. Surveillance

To determine the drivers of disease occurrence and to work toward designing effective control strategies, there is a need to identify pathogens that are endemic in various aquatic ecosystems used for fish farming. Surveillance is useful for the early detection of etiological agents of disease as well as for determining existing pathogens and their prevalence in farmed fish and wild reservoirs. Surveillance is also useful in determining the epidemiological patterns of endemic viral diseases in different ecosystems. It is also useful in identifying endemic hotspots and antigenic variants needed to guide vaccine design [3,109,110,111,112,113]. Moreover, a good understanding of pathogens endemic in each ecosystem is useful for the design of protective vaccines against prevailing diseases in the area. Where possible, the application of surveillance guidelines established by the Aquatic Animal Health Code of the World Organization of Animal Health (OIE) [114] should be used. For transboundary diseases, transborder surveillance programs can be established using common standard diagnostic tools across different countries.

### 3.4. Eradication of Fish Diseases in Aquaculture

Eradication refers to eliminating the infectious agent from a given area or reducing its presence to insignificantly low levels [115]. Disease elimination involves reducing the reproductive number (R_0_) of the pathogen to zero. Major factors to be considered in the initial assessment of the disease status before embarking on an eradication program include determining the incidence, transmission index, and existing control measures [115]. For an eradication program to be successful, elimination measures should aim at reducing the incidence and transmissibility (R_0_) to zero. In marine aquaculture, only a few reports of disease eradication by total elimination have been documented [116,117]. During the period of 1998–1999, the ISAV epidemic in Scotland was controlled by eradication [116]. Even though measures were established to prevent the resurgence of ISAV in the area, the disease re-emerged in 2008 and 2009, though it was limited to the Shetland Islands [117]. Another example is the elimination of VHSV in Denmark. During the period between 1960 and 2010, Denmark embarked on an eradication program that resulted in the elimination of VHSV from Danish farms [118]. The major threat to eradication approaches is that wild reservoirs of the disease may not be easily eliminated. Despite this, eradication is an effective disease control strategy in places where pathogens can be eliminated from areas used for aquaculture.

### 3.5. Selective Breeding for Genetic Resistance

The outcome of infections is influenced, among others, by host factors. Inherent resistance to infection has been documented in some fish species, which has given direction in scientific research to determine genetic markers associated with disease resistance. One of the methods commonly used for identifying disease-resistant traits in fish is the use of marker-assisted selection (MAS) based on detecting the quantitative trait loci (QTL) for disease resistance [119]. A successful example of QTL analyses applied to selective breeding is the case of IPNV resistance in Atlantic salmon, in which a major QTL has contributed to a significant reduction in IPNV cases in the salmon industry in Norway [120,121,122]. This shows that QTL-based selection of disease-resistant fish can be useful in reducing the occurrence of viral diseases in aquaculture. Several studies have been carried out to determine the QTLs for major viral diseases of farmed fish (Table 1).

Apart from QTL analysis, genetic variability in different immune genes has been used as a molecular marker associated with polymorphism and resistance/susceptibility to disease in different vertebrate species [137]. In fish, specific MHC alleles of class I and II genes have been linked to disease resistance in different fish species (Table 2).

Other genes associated with disease resistance in fish include toll-like receptors (TLRs), melanoma differentiation-associated protein 5 (MDA5), IL-10β, retinoic-acid-inducible gene I (RIG-I), laboratory of genetics and physiology 2 (LGP2), and cadherin-1 (CDH1) (Table 2). The selection of disease-resistant fish has the potential to reduce viral infections in aquaculture.

### 3.6. Antiviral Compounds

Since 1963, when idoxuridine was approved as the first antiviral drug, several other antiviral drugs have been approved for the treatment of viral diseases in higher vertebrates [147]. By 2016, 90 antiviral drugs [147] had been approved for human clinical use. Studies carried out using different fish species and cell lines show that several antiviral compounds have the potential to serve as therapeutic drugs against different viruses infecting fish (Table 3).

There is a potential for some of these drugs to reduce viral infections in farmed fish, but the approval process will be a challenge. Another approach is the use of probiotics having antiviral properties. Probiotics are microorganisms with variable properties, such as enhancing growth and increasing host immunity against various pathogens. Recent developments have shown that some probiotics have antiviral properties. For example, *Bacillus subtilis* produces surfactin, an antiviral compound against various mammalian and avian viruses, such as porcine parvovirus (PPV), pseudorabies virus (PRV), and infectious bursal disease virus (IBDV) [166]. It also has antiviral properties against fish viruses such as VHSV [161]. Other probiotics shown to have antiviral properties against fish viruses are shown in Table 4. Altogether, antiviral drugs and probiotics have the potential to reduce the occurrence of viral diseases in aquaculture.

### 3.7. Vaccine Development and Immunization Strategies

Fish vaccination is the most important environmentally friendly disease control strategy that has not only contributed to the prevention and control of viral diseases but has also contributed to a reduction in the use of antibiotics in aquaculture. Various reviews have reported the status and progress in vaccine development in aquaculture in recent decades [176,177,178,179,180,181,182,183,184,185]. These studies have brought to light several shortcomings that have paved the way to current research in fish vaccinology. Although inactivated whole viral vaccines account for the largest proportion of vaccines used in aquaculture (Table 5), the level of protection attained by these vaccines is equivocal [186].

Therefore, the challenge has been to develop replicative vaccines as alternatives. However, efforts to produce live attenuated vaccines have been marred by the fear of reversion to virulence [192]. Moreover, concerns that live vaccines could be pathogenic to other organisms found in aquatic environments used for fish farming hinder efforts to develop them. As an alternative approach, some studies have focused on developing DNA vaccines [193,194,195]. To date, two DNA vaccines have been licensed for use in aquaculture, one against IHN in Canada [194,196], while the other is against pancreas disease (PD) in Norway [197]. Most fish vaccines are administered by injection, which is labor intensive if done by human vaccinators, or alternatively, the use of automated vaccination machines, which requires considerable investment. Injection vaccination of fish induces stress that may result in temporary immunosuppression. Oral vaccination, on the other hand, where vaccines are administered through feed, is more preferred as vaccines can be administered without stressing the fish. However, there are few examples of oral vaccines being efficacious, although some have been licensed for use in aquaculture [187].

Other efforts have focused on identifying correlates of protective immunity for use as benchmarks in vaccine development [99,198,199], optimizing antigen dose used in vaccine formulations [99], selection of highly immunogenic antigens [200], optimizing challenge models for vaccine efficacy trials [201,202,203], developing multivalent vaccines against multipathogens [187], optimizing prime-boost vaccination regimes [204,205], and identifying potent adjuvants with few side effects [206,207,208]. These efforts have contributed to improving the quality of current commercial vaccines used in aquaculture, ultimately contributing to a reduction in the prevalence of viral diseases in mariculture.

### 3.8. Management and Husbandry Practices

Good management practices are critical to minimize stress and reduce disease occurrence. Intensive fish farming systems often lead to high stocking densities, which induce stress in fish and increase the disease transmission index (>R_0_) due to high contact between fish [15]. Therefore, it is vital that optimal stocking densities are used. Other stressors such as poor feeding regimes and the use of feed with a negative impact on feed conversion should be avoided. Apart from good management practices on individual fish farms, the adoption of “area-based management plans” has proved to be effective in reducing disease occurrences in most countries [209,210,211]. Key elements of “area-based management” include the use of agreed, good husbandry practices, synchronous fallowing, separation of generations, and harvesting protocols for all facilities within the managed area [211]. Other factors include adherence to official disease control policies, carrying out routine inspection on health status of stock, use of the same vaccines and vaccination regimes, use of common biosecurity plans, adherence to agreed fallowing timelines and protocols, adherence to agreed stocking densities, as well as use of agreed protocols on the disposal of dead fish.

Other management practices contributing to the control of viral diseases in marine aquaculture include zoning and the use of standard siting distances between fish farms, shown to be useful in reducing the transmission of viruses such as ISAV, SAV, and IPNV in Chile, Scotland, and Norway [212,213,214,215,216,217]. Jarp and Karlson [218] and McClure et al. [216] noted that distances >5 km between farm sites were effective in reducing ISAV transmission in Norway and Canada, respectively. The Norwegian Food Safety Authority enforces a 5 km restriction zone and a 10 km observation zone around ISAV-infected farms [219]. Demarcating the Norwegian coastline into the north and south administrative units has proved to be an efficient barrier separating the endemic south region from the disease-free north region of SAV [220].

An upcoming technology having the potential to reduce the transmission of marine viruses is the recirculation aquaculture system (RAS). This technology is based on the culture of marine fish species using an in-land controlled RAS environment. Given that there is no direct interaction between farmed and wild fish, the risk of infection from wild to farmed fish is prevented. Additionally, the chances of disease spillover from farmed fish into the environment are also minimal. Although expensive, this technology has the potential to reduce disease transmission in aquaculture.

### 3.9. Disease Control Polices

Disease control policies are principles, plans, or courses of action pursued by governments, organizations, and individual fish farmers aimed at preventing or reducing disease occurrence [221]. Several documents have been developed by different international organizations and countries to deal with disease control policies [221]. Figure 2 shows a structure involving global agencies, multinational, national, and farm-level organizations involved in the implementation of disease control policies. At global level, the OIE is an intergovernmental body responsible for the implementation of fish disease control policies. It has developed two international standards, namely, the aquatic animal health code that enlists all major diseases and the manual of diagnostic tests for aquatic animals [222]. It is mandatory to report all occurrences of diseases listed in the aquatic animal health code by member countries to the OIE [222]. Apart from the OIE, multinational unions develop fish health policies aimed at ensuring that all member states use the same disease control strategies. For example, the European Union (EU) has a fish health code through which EU member states agree on the list of pathogens to control using agreed prevention and control measures [223]. In some cases, multinational unions form intercountry teams responsible for disease monitoring and research. Such undertakings allow the exchange of information and technical expertise.

National disease control policies must include preventing the introduction of exotic pathogens, early disease detection, monitoring health status, certification of export of live fish and fish products, movement restriction, and zoning [223]. All health certificates must be signed by authorized experts. In addition, national policies should include statutory regulations on the inspection of fish farms, sampling and testing programs, outbreak investigations, and enforcement of disease control measures. National policies are also responsible for ensuring that all relevant records are kept for stipulated durations. For example, for a disease traceback system to be effective, movement records of live fish and fish products must be kept for several years for accessibility by fish health inspectors. Similarly, fish health and mortality records should be kept for several years.

## 4. Conclusions

Marine fish farming has great potential for improving human livelihood as well as making a significant contribution to the success of the Sustainable Development Goals (SDGs) of the United Nations. To achieve this, there is a need to overcome several challenges hindering success in the control and prevention of viral infections causing high economic losses in farmed fish. As shown in this synopsis, a multifaceted approach involving several strategies, such as vaccine development and immunization, potential use of antiviral drugs, selective breeding for disease resistance, surveillance, policy formulation, and implementation of biosecurity measures is bound to offer a more effective long-lasting solution.

## Figures and Tables

**Figure 1 pathogens-10-00673-f001:**
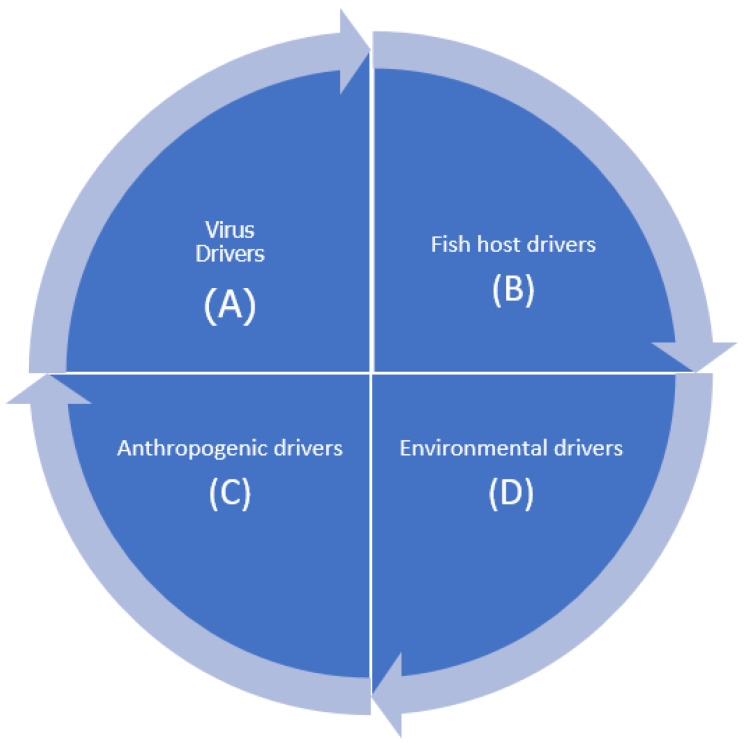
Drivers of emerging infectious diseases in aquaculture. (**A**) Virus drivers include mutations leading to virulent strains and the ability to infect different host species. (**B**) Fish host drivers include susceptibility to infection due to low immunity. (**C**) Anthropogenic drivers include farming practices that include high stocking density and poor feeding regimes that cause fish to become stressed and increase susceptibility to infection. (**D**) Environmental drivers include adverse conditions such as changes in alkalinity, reduced dissolved oxygen, and increase in nitrites that predispose fish to viral infection.

**Figure 2 pathogens-10-00673-f002:**
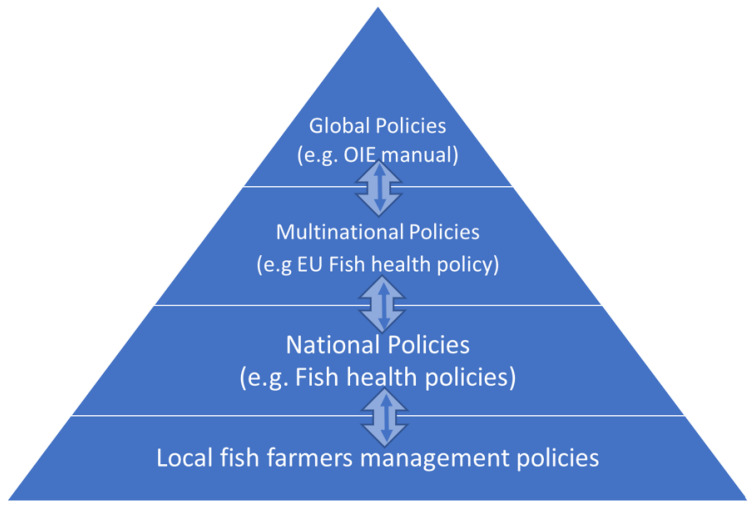
Disease control policies showing different levels namely, global policies involving institutions such as the OIE, multinational polices such as the European Union (EU), national polices, and local farmers’ management plans. Arrows show that policies developed at local fish farmer level can be adopted to become part of the national fish health policy. Similarly, national fish health policies can be adopted and included in multinational and global policies by organizations such as the OIE. The converse also applies: policies developed at global level can be disseminated for inclusion at multinational, national, and local fish famer levels.

**Table 1 pathogens-10-00673-t001:** Disease resistance traits determined by quantitative trait loci.

Disease/Pathogen	ABRREV	Fish Species (Scientific Name)	References
Infectious salmon anemia	ISAV	Atlantic salmon (*Salmo salar* L.)	[123]
Salmonid alphavirus	SAV	Atlantic salmon (*Salmo salar* L.)	[124]
Viral hemorrhagic septicemia virus	VHSV	Rainbow trout (*Oncorhynchus mykiss*)	[125]
Viral hemorrhagic septicemia virus	VHSV	Turbot (*Scophthalmus maximus*)	[126]
Infectious pancreatic necrosis virus	IPNV	Atlantic salmon (*Salmo salar* L.)	[120,127]
Infectious pancreatic necrosis virus	IPNV	Rainbow trout (*Oncorhynchus mykiss*)	[128,129]
Lymphocystis disease	LCDV	Japanese flounder (*Paralichthys olivaceus*)	[130]
Salmon alphavirus	SAV	Atlantic salmon (*Salmo salar* L.)	[131,132]
Red sea bream iridovirus disease	RSBIV	Sea Bream (*Pagrus major*)	[133]
Nervous necrosis virus	NNV	Asian sea bass (*Lates calcarifer*)	[134]
Infectious hematopoietic necrosis virus	IHNV	Rainbow trout (*Oncorhynchus mykiss*)	[135]
Piscine cardiomyopathy syndrome virus	PCMSV	Atlantic salmon (*Salmo salar* L.)	[136]

**Table 2 pathogens-10-00673-t002:** Immune markers of disease resistance in different fish species.

Gene	Disease/Pathogen	ABBR	Fish Species (Scientific Name)	Ref.
MHC-I and II	Infectious anemia virus	ISAV	Atlantic salmon (*Salmo salar* L.)	[123]
MHC-I and II	Infectious anemia virus	ISAV	Atlantic salmon (*Salmo salar* L.)	[123]
MHC-II	Infectious hematopoietic necrosis virus	IHNV	Rainbow trout (*Oncorhynchus mykiss*)	[138]
MHC-Ia	Infectious hematopoietic necrosis virus	IHNV	Rainbow trout (*Oncorhynchus mykiss*)	[139]
MHC-II	Infectious hematopoietic necrosis virus	IHNV	Cutthroat trout (*Oncorhynchus clarkii*)	[140]
IL-10β	Cyprinid herpesvirus	CyHV-3	Common carp (*Cyprinus carpio*)	[141]
TLRs	Cyprinid herpesvirus	CyHV-3	Common carp (*Cyprinus carpio*)	[142]
TLR3	Grass carp reovirus	GCRV	Grass carp (*Ctenopharyngodon idella*)	[143]
TLR22	Grass carp reovirus	GCRV	Grass carp (*Ctenopharyngodon idella*)	[144]
MDA5	Grass carp reovirus	GCRV	Grass carp (*Ctenopharyngodon idella*)	[145]
RIG-I	Grass carp reovirus	GCRV	Grass carp (*Ctenopharyngodon idella*)	[145]
LGP2	Grass carp reovirus	GCRV	Grass carp (*Ctenopharyngodon idella*)	[146]

**Table 3 pathogens-10-00673-t003:** Antiviral compounds and probiotics tested against viruses infecting marine fish.

Antiviral Compounds	Viral Pathogens	Abbrev	Ref.
Flavonoids	Viral hemorrhagic septicemia virus	VHSV	[148]
Flavonoids	Infectious hematopoietic necrosis virus	IHNV	[148]
Amantadine	Infectious hematopoietic necrosis virus	IHNV	[149]
Dextran	Infectious pancreatic necrosis virus	IPNV	[150]
Dextran	Infectious hematopoietic necrosis virus	IHNV	[150]
Gymnemagenol	Nervous necrosis virus	NNV	[151]
Dasyscyphin C	Grouper iridovirus	GIV	[152]
Casein	Infectious hematopoietic necrosis virus	IHNV	[153]
Acyclovir	Herpesvirus salmonis	HPV	[154]
Brivudin (BVDU)	Herpesvirus salmonis	HPV	[155]
Coumarin	Spring viremia of carp virus	SVCV	[156]
Saikosaponin D	Spring viremia of carp virus	SVCV	[157]
Arctigenin derivatives	Spring viremia of carp virus	SVCV	[158]
Arctigenin derivatives	Infectious hematopoietic necrosis virus	IHNV	[159]
Dasyscyphin C	Nervous necrosis virus	NNV	[160]
Surfactant	Viral hemorrhagic septicemia virus	VHSV	[161]
Honokiol and moroxydine hydrochloride	Grass carp reovirus	GCRV	[162,163]
Lipophilic thiazolidine derivatives (LJ001, JL118, JL122)	Viral hemorrhagic septicemia virus	VHSV	[164]
Lipophilic thiazolidine derivatives (LJ001, JL118, JL122)	Infectious hematopoietic necrosis virus	IHNV	[164]
Lipophilic thiazolidine derivatives (LJ001, JL118, JL122)	Spring viremia of carp virus	SVCV	[164]
furan-2-yl acetate	Nodavirus	NNV	[165]

**Table 4 pathogens-10-00673-t004:** Probiotics tested against viruses infecting marine fish.

Probiotics	Viral Pathogens	Abbrev	Ref.
Aeromonas species	Infectious hematopoietic necrosis virus	IHNV	[167]
Pseudomonas spp.	Infectious hematopoietic necrosis virus	IHNV	[167]
Corynebacterium species	Infectious hematopoietic necrosis virus	IHNV	[167]
Bacillus subtilis	Viral hemorrhagic septicemia virus	VHSV	[161]
Lactobacillus	Lymphocystis disease virus	LCDV	[168]
*Aeromonas hydrophila* strains M-26 and M-38	Infectious hematopoietic necrosis virus	IHNV	[169]
*V. alginolyticus* strain V-23	Herpesvirus salmonis	HPV	[169]
*Bacillus subtilis* E20	Singapore grouper iridovirus	SGIV	[170]
*Lactobacillus plantarum*	Singapore grouper iridovirus	SGIV	[171]
*Saccharomyces cerevisiae* P13	Singapore grouper iridovirus	SGIV	[172]
*B. subtilis 7K*	Singapore grouper iridovirus	SGIV	[173]
*Shiwanella* spp. *strain 0409*	Betanodvirus	NNV	[174]
*Clostridium butyricum* (Cb)	Gibel carp herpesvirus	CaHV	[175]

**Table 5 pathogens-10-00673-t005:** Licensed vaccines developed for marine fish species.

Disease	Major Fish Host	Vaccine Type	Delivery	Country	Reference
Infectious hematopoietic necrosis	Salmonids	DNA	IM	Canada	[176,187]
Infectious pancreatic necrosis	Salmonids, sea bass, sea bream, turbot, Pacific cod	Inactivated	IP	Norway, Chile, UK	[176,187]
		Subunit	Oral	Canada, USA	[176,187]
		Subunit	IP	Canada, Chile, Norway	[176,187]
Infectious salmon anemia	Atlantic salmon	Inactivated	IP	Norway, Chile, Ireland, Canada	[176,187]
Infectious salmon anemia	Atlantic salmon	Subunit	IP	Norway, Chile, Ireland, Canada	[176,187]
Infectious spleen and kidney necrosis	Asian seabass, grouper, Japanese yellowtail	Inactivated	IP	Singapore	[176]
Red seabream iridovirus	Red seabream	Inactivated	IP	Japan/South Korea	[188]
Viral hemorrhagic septicemia virus	Olive flounder	Inactivated	IP	South Korea	[188]
Nervous necrosis virus	Grouper	Inactivated	IP	Japan	[189,190]
Pancreas disease virus	Salmon	DNA	IM	Norway, Chile, UK	[176,187]
Pancreas disease virus	Salmon	Inactivated	IP	Ireland	[191]

## Data Availability

Not applicable.

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
