# Peer review of "Challenges and Solutions to Viral Diseases of Finfish in Marine Aquaculture"

_pathogens, 2021, doi:10.3390/pathogens10060673_

Round 1

Reviewer 1 Report

In Figure 1 and 2; There is little information and it is difficult to understand the contents. of Figs. Information should be enriched in the figure itself or in the figure legends.

In Table 5; Please fill in the blank of "vaccine type".

Author Response

Response to reviewer-2

QUERY In Figure 1 and 2; There is little information and it is difficult to understand the contents. of Figs. Information should be enriched in the figure itself or in the figure legends.

RESPONSE: We have added more information on the figure legends as recommended.

QUERY In Table 5; Please fill in the blank of "vaccine type".

RESPONSE: We have inserted the missing information Table 5.

Reviewer 2 Report

This is a review paper covering a wide range of challenges caused by viral diseases in aquaculture of fish. It is a well-written paper and the amount of literature covered is impressive. While reading, it brought my attention to many papers I was not aware of. The manuscript is a bit different than most other reviews on viral diseases in aquaculture since it puts together so many topics. However, this makes the manuscript very broad and some of the topics are covered in a superficial way and not really in-depth.

More specific comments

Line 33: I am missing a reference after this sentence “Among these are viral diseases most of which cause high economic losses”

It is also important to remember and highlight that viral diseases cause reduced fish welfare. I think the fish welfare perspective should be emphasized more in the introduction or elsewhere in the text.

Line 99-101: Is ref 31 cited correctly? In the abstract it says that the prevalence of these viral pathogens is very low in wild salmonids. This has also been shown in later publications.

Line 134: VERV is used here and NNV is used in line 114 and line 140/142 – same virus?

Line 149: Ref 49 is modelling environmental effects of aquaculture with focus on eutrophication. Is this the correct use of this ref?  “As such, they also serve as viral transmission sites between farmed  and wild fish [49]”

Line 203: “The traditional approach of isolating viruses from infected tissues by cell culture followed by virus identification using different techniques is the most common approach for the diagnosis of fish viral diseases.” I think cell culture isolation of virus is not common any longer in diagnostics. qPCR detection is the most common method, but has to be combined with other detection methods to set an official diagnosis (commonly immunohistochemistry). Limitations in diagnostic tools should focus on the limitaions of qPCR detection, which does not necessarily detect live virus and other limitations.

Line 380: should refer to table 3 and not table 4.

P 29-46: What is this supposed to show? Something has occurred to the layout of this part of the manuscript.

Author Response

Response to reviewer-1

QUERY Line 33: I am missing a reference after this sentence “Among these are viral diseases most of which cause high economic losses”

RESPONSE: References have now been added, see line 34 (References 2, 3 and 4)

QUERY: It is also important to remember and highlight that viral diseases cause reduced fish welfare. I think the fish welfare perspective should be emphasized more in the introduction or elsewhere in the text.

RESPONSE: The impact of viral diseases in reducing fish welfare has now been included in the introduction (See lines 34 to 37)

QUERY Line 99-101: Is ref 31 cited correctly? In the abstract it says that the prevalence of these viral pathogens is very low in wild salmonids. This has also been shown in later publications.

RESPONSE: Yes reference 38, formerly reference 31, is correctly cited. While we agree with the reviewer that the prevalence of these pathogens is low in wild salmonids, we found it necessary to mention the risk posed by farmed salmon in transmitting viruses to wild fish.

QUERY Line 134: VERV is used here and NNV is used in line 114 and line 140/142 – same virus?

RESPONSE: VERV has been replaced with NNV because it is the same virus (see line 139)

QUERY Line 149: Ref 49 is modelling environmental effects of aquaculture with focus on eutrophication. Is this the correct use of this ref?  “As such, they also serve as viral transmission sites between farmed and wild fish [49]”

RESPONSE: Reference 49 has been deleted and the sentence has been modified to imply that “they are bound to serve as virus transmission sites between farmed and wild fish”. (see lines 155-156)

QUERY Line 203: “The traditional approach of isolating viruses from infected tissues by cell culture followed by virus identification using different techniques is the most common approach for the diagnosis of fish viral diseases.” I think cell culture isolation of virus is not common any longer in diagnostics. qPCR detection is the most common method, but has to be combined with other detection methods to set an official diagnosis (commonly immunohistochemistry). Limitations in diagnostic tools should focus on the limitaions of qPCR detection, which does not necessarily detect live virus and other limitations.

RESPONSE: In agreement with the reviewer we have added lines 216 -223 explaining that PCR has become the most common diagnostic tools. Limitation regarding the use of PCR have been included as recommended (lines 216 -223).

QUERY Line 380: should refer to table 3 and not table 4.

RESPONSE: Correction done as recommended. As a matter the numbering of all Table has been corrected (see lines 375, 379, 382, 393, 402, and 413)

P 29-46: What is this supposed to show? Something has occurred to the layout of this part of the manuscript.

RESPONSE: This is an editorial formatting error from our submitted manuscript into the journal format (Pathogens mdpi), we hope it will corrected in the final version.

Response to reviewer-2

QUERY In Figure 1 and 2; There is little information and it is difficult to understand the contents. of Figs. Information should be enriched in the figure itself or in the figure legends.

RESPONSE: We have added more information on the figure legends as recommended.

QUERY In Table 5; Please fill in the blank of "vaccine type".

RESPONSE: We have inserted the missing information Table 5.
